# MODEL SELECTION OF ANOMALY DETECTORS IN THE ABSENCE OF LABELED VALIDATION DATA

## ABSTRACT

Anomaly detection requires detecting abnormal samples in large unlabeled datasets. While progress in deep learning and the advent of foundation models has produced powerful unsupervised anomaly detection methods, their deployment in practice is often hindered by the lack of labeled data—without it, the detection accuracy of an anomaly detector cannot be evaluated reliably. In this work, we propose a general-purpose framework for evaluating image-based anomaly detectors with synthetically generated validation data. Our method assumes access to a small support set of normal images which are processed with a pre-trained diffusion model (our proposed method requires no training or fine-tuning) to produce synthetic anomalies. When mixed with normal samples from the support set, the synthetic anomalies create detection tasks that compose a validation framework for anomaly detection evaluation and model selection. In an extensive empirical study, we find that our synthetic validation framework selects the same models and hyper-parameters as selection with a ground-truth validation set when evaluated on natural images (i.e., images of flowers and birds). We also find that prompts selected by our method for CLIP-based anomaly detection outperforms all other prompt selection strategies and leads to the overall best detection accuracy on all datasets, including the challenging MVTec-AD dataset.

## 1 INTRODUCTION

Anomaly detection, automatically identifying samples that deviate from normal behavior, is an important technique for supporting medical diagnosis (Fernando et al., 2021), safeguarding financial transactions (Ahmed et al., 2016), bolstering cybersecurity (Mirsky et al., 2018; Siadati & Memon, 2017), and ensuring smooth industrial operations (Bergmann et al., 2019). There has been significant progress in data-driven approaches for unsupervised anomaly detection (Deecke et al., 2021; Liznerski et al., 2022; Jeong et al., 2023b; Li et al., 2023a; 2021; Reiss et al., 2021). Before an anomaly detection method can be safely deployed in a new application, one must trust that it performs as well as expected, but performing such an evaluation is often hindered by a major barrier: the absence of labeled validation data. In many applications, validation data is typically absent because anomalies are rare and the large volume of available data is too expensive to label (Görnitz et al., 2013; Trittenbach et al., 2021).

Labeled validation data is also beneficial for zero-shot anomaly detection. With recent developments in foundation models, it has become possible to pre-train a large model on large-scale data from one domain and then to deploy it for a new anomaly detection task. Specifically, CLIP-based anomaly detection approaches (Jeong et al., 2023b; Liznerski et al., 2022; Zhou et al., 2021; Esmaeilpour et al., 2022) have shown great performance in a variety of domains. While these approaches provide the exciting possibility to construct new anomaly detectors for new applications on demand, they could be more readily deployed in real-world applications if labeled validation data could aid with prompt selection and model evaluation.

In this work, we study the efficacy of synthetically generated anomalies for model selection of image-based anomaly detectors. Given a new image-based anomaly detection task, such as detecting faulty objects in a manufacturing plant, we assume access to a small set of normal samples (Zhao et al., 2021). Our approach leverages diffusion models (Ho et al., 2020; Song et al., 2021; Jeong et al., 2023a) to generate synthetic anomalies from this small set of examples. The synthetic anoma-

lies are then mixed with the normal examples to provide a synthetic validation set. In extensive experiments, ranging from natural images to industrial applications, we show, that the performance on anomaly detection benchmarks with synthetically generated validation sets often matches results on a validation set with ground truth labels.

Importantly, we develop a method for anomaly generation, that does not require training or fine-tuning a new diffusion model for the specialized anomaly detection task. Instead, we use the available normal samples and a diffusion model pre-trained on ImageNet (Deng et al., 2009; Dhariwal & Nichol, 2021) to interpolate between those normal samples (Jeong et al., 2023a). We find that even for domains that are far from ImageNet, such as the industrial images in the MVTec-AD dataset (Bergman & Hoshen, 2020), this scheme generates synthetic anomalies with realistic backgrounds and consistent visual patterns.

Our work makes the following contributions:

- In Sec. 3.1, we present a framework for selecting image-based anomaly detection models based on synthetically generated anomalies. Figure 1 shows the outline of our approach.
- We propose a practical technique for generating synthetic anomalies that uses a general-purpose pre-trained diffusion model—without any fine-tuning or auxiliary datasets, described in Sec. 3.2.
- We empirically evaluate our method on a wide range of anomaly detection tasks and demonstrate its success on two use-cases: model selection amongst candidate anomaly detectors (Sec. 4.2) and prompt selection for anomaly detection with CLIP (Sec. 4.3).

## 2 RELATED WORK

**Unsupervised anomaly detection.** Recent advances in anomaly detection models include autoencoder-methods (Chen & Konukoglu, 2018; Principi et al., 2017), deep one-class classification (Ruff et al., 2018; 2019), and self-supervised learning-based methods (Bergman & Hoshen, 2020; Hendrycks et al., 2019b; Qiu et al., 2021; Sohn et al., 2020; Qiu et al., 2022a; Schneider et al., 2022). While these methods are unsupervised, their architectures and training frameworks involve various hyperparameters, which can have a strong impact on detection accuracy (Campos et al., 2016; Goldstein & Uchida, 2016; Han et al., 2022). While Ding et al. (2022) propose to circumvent model selection by building an ensemble of candidate methods, model selection typically requires labeled validation data.

In outlier exposure (Hendrycks et al., 2019a), which has been extended to anomaly detection with contaminated data (Qiu et al., 2022b) and to meta-learning (Li et al., 2023b), further improvements in detection accuracy come from using samples from an auxiliary dataset, usually Tiny-ImageNet. Although these auxiliary samples provide valuable additional training signal, they are too dissimilar from normal samples and can be easily detected; therefore, they are not useful for model evaluation or selection. Work in semi-supervised anomaly detection (Görnitz et al., 2013; Das et al., 2016; Trittenbach et al., 2021; Li et al., 2023a) assumes access to a training set with some labeled anomalies but obtaining such data is unrealistic in most real-world applications.

**Anomaly detection with foundation models.** Nowadays, large models can be pre-trained on massive datasets to learn rich semantic image features which have proven useful for anomaly detectors in other vision domains—little to no additional training required. Large vision models such as vision transformers (ViT, Dosovitskiy et al. (2021)) or residual networks (ResNet, He et al. (2016)) pre-trained on ImageNet (Deng et al., 2009), can often be used as anomaly detectors in other vision domains, either without fine-tuning or by fine-tuning an outlier exposure objective (Deecke et al., 2021; Fort et al., 2021; Mirzaei et al., 2023; Reiss et al., 2021).

Another powerful class of foundation models are vision-language models like CLIP (Radford et al., 2021). Esmaeilpour et al. (2022) use a support set of normal samples to learn a text description generator for CLIP-based out-of-distribution detection. With hand-crafted text prompts CLIP can be employed on a new anomaly detection (Liznerski et al., 2022) or anomaly segmentation task (Jeong et al., 2023b; Zhou et al., 2021) without any training, i.e. in a zero-shot manner, which means no training data for the new task is required. However, detection performance depends on several key hyperparameters, especially the choice of prompts (Liznerski et al., 2022; Jeong et al., 2023b), which implies that labeled validation data for prompt selection would be highly beneficial. However,

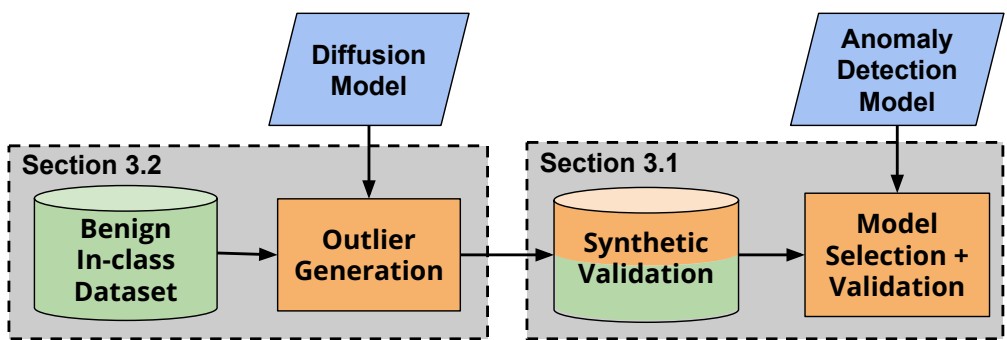

Figure 1: We propose to use a pretrained diffusion model to turn a small dataset of normal samples into a synthetic validation set containing both real normal samples and synthetic anomalies, described in Sec. 3.2. The synthetic validation set is then used for model selection and validation, as described in Sec. 3.1. Components in blue are frozen, components in green are real data, and components in orange are our techniques.

in the age of foundation models, when task-specific instances of foundation models require little to no data, it does not make sense to assume access to labeled validation data.

**Evaluation of tabular and time-series anomaly detection.** Nguyen et al. (2016); Marques et al. (2015; 2020) propose unsupervised model selection strategies with so called "internal metrics" that can be computed from predicted anomaly scores on unlabeled data. (See Ma et al. (2023) for a review.) Meta-training can offer another set of approaches for unsupervised anomaly detector selection (Schubert et al., 2023; Zhao et al., 2021; 2022). However, since meta-learning requires a large number of related labeled datasets, their application has been limited to tabular data, and prior work on internal metrics has been applied to tabular anomaly detection only (Nguyen et al., 2016; Marques et al., 2015; 2020; Ma et al., 2023). In contrast, we exploit recent advances in diffusion models to enable anomaly detection model selection without labeled validation data in the vision domain. In time-series anomaly detection, model evaluation is also known to be biased and unreliable due to poorly labeled test data (Wu & Keogh, 2020; Wagner et al., 2023). Here, hand-crafted anomalies can be useful (Lai et al., 2021). But for the evaluation of vision-based anomaly detectors, synthetic anomalies need to go beyond pixel-wise manipulation and are instead required to reflect abnormal patterns in a more complex, and often semantic, level of abstraction.

**Generating images with guidance.** Diffusion models (Ho et al., 2020; Song et al., 2021) have recently been found to outperform other generative models (Dhariwal & Nichol, 2021). Although generative models are traditionally used to generate in-distribution data, a variety of prior work has proposed techniques that guide generative models to generate new distributions. Specifically, text-guided generation (Kim et al., 2022; Kwon et al., 2023; Kawar et al., 2023; Mokady et al., 2023; Tumanyan et al., 2023) with CLIP (Radford et al., 2021) guided the diffusion-based image generation process with text prompts to generate samples from a distribution of interest, e.g. "cat with glasses". Luo et al. (2023) use CLIP embeddings of the labels of different classes to guide StyleGAN (Karras et al., 2019; 2020) to generate images for the evaluation of image classifiers. Similarly, Jain et al. (2023) use CLIP for model diagnosis. These approaches require the target labels (e.g, gender, glasses, etc.) as inputs, but the nature of anomalies is typically unknown. Instead, we rely on DiffStyle (Jeong et al., 2023a), which provides training-free guidance for image generation by interpolating two images using the reverse DDIM process (Song et al., 2021). We find that interpolating between two normal samples will typically preserve the dominant visual features, such as realistic textures and background, while introducing slight manipulations that make the generated images promising candidates for synthetic anomalies in our proposed model selection framework.

## 3 METHOD

In this section, we propose to use existing diffusion-based image generation techniques to generate synthetic anomalies. By using these synthetic anomalies as a *synthetic validation dataset*, we enable

model selection in domains where a real validation dataset does not exist. In Sec. 3.1, we describe how synthetic anomalies can be used for model selection. We then propose a synthetic anomaly generation approach in Sec. 3.2. Figure 1 demonstrates the overall process used by our method.

## 3.1 MODEL SELECTION WITH SYNTHETIC ANOMALIES

The absence of labeled validation data is a major roadblock in the deployment of anomaly detection methods. However, normal data can usually be obtained. For this reason, we follow Zhao et al. (2021), and assume access to a set of normal samples we call the *support set* $X_{\text{support}}$. In our empirical study, we show that the support set can have as few as 10 normal samples. Using this support set, we wish to construct a synthetic validation set that can aid model selection.

Creating a synthetic validation set and using it for model selection entails the following steps:

**Step 1: Partitioning the support set.** The support set $X_{\text{support}}$, is randomly partitioned into style images $X_{\text{style}}$, content images $X_{\text{content}}$, and normal validation images $X_{\text{in}}$. $X_{\text{style}}$ and $X_{\text{content}}$ are used for anomaly generation, and $X_{\text{in}}$ is held out for evaluation.

**Step 2: Generating synthetic anomalies.** The style and content images are processed with Diff-Style (Jeong et al., 2023a) to produce synthetic anomalies $\tilde{X}_{\text{out}}$. Details will be given in Sec. 3.2.

**Step 3: Mixing the synthetic validation set.** The normal validation images $X_{\text{in}}$ are combined with the synthetic anomalies $\tilde{X}_{\text{out}}$ to produce a labeled synthetic validation set,

$$\mathcal{D} = \{(x, 1) | x \in \tilde{X}_{\text{out}}\} \cup \{(x, 0) | x \in X_{\text{in}}\}, \tag{1}$$

where the label 1 indicates an anomaly and 0 a normal image.

**Step 4: Evaluating detection accuracy of candidate models.** In a final step, candidate models are evaluated in terms of their detection accuracy on the synthetic validation set $\mathcal{D}$. Anomaly detection methods are typically evaluated in terms of AUROC, the area under the receiving operator characteristic curve (Emmott et al., 2015; Maxion & Tan, 2000).

Since we assume a small support set, training or fine-tuning a generative (diffusion) model is infeasible. Instead, in Sec. 3.2, we propose to use a DDIM (Song et al., 2021) pre-trained on ImageNet and DiffStyle, a training-free method for diffusion-based image-to-image style transfer, and adapt it for generating synthetic anomalies. Our method does not perform any model training or fine-tuning and does not require any data outside of the support set.

## 3.2 GENERATING SYNTHETIC ANOMALIES

We use DiffStyle (Jeong et al., 2023a) to generate synthetic anomalies with a pretrained DDIM. From style images $X_{\text{style}}$, content images $X_{\text{content}}$, DiffStyle takes any style-context images pair as input—a "style image" $I^{(1)}$ and a "content image" $I^{(2)}$—and produces a new, generated image with $I^{(2)}$'s content and $I^{(1)}$'s style. To achieve this, $I^{(1)}$ and $I^{(2)}$ are mapped into the diffusion model's latent space through the forward diffusion process, producing latent vectors $x_T^{(1)}$ and $x_T^{(2)}$. We refer to the h-space (i.e., the inner-most layer of the UNet) of $x_T^{(1)}$ and $x_T^{(2)}$ as $h^{(1)}$ and $h^{(2)}$ respectively. The h-space has been shown to be a meaningful semantic space in the image domain, enabling properties such as linearity and composition between images and can be manipulated during a diffusion model's image generation process to achieve desired properties. We refer readers for more details related to h-space to Kwon et al. (2023) and Jeong et al. (2023a).

Given two latent vectors $h^{(1)}$ and $h^{(2)}$, a simple linear interpolation is performed to style-transfer between two images: $h^{(\text{gen})} = (1 - \gamma)h^{(1)} + \gamma h^{(2)}$ where $\gamma$ represents the relative strength of the content image during style transfer[1]. We use $\gamma = 0.7$ as the default in our experiments. We then perform the asymmetric reverse diffusion process using $x_T^{(1)}$, replacing the h-space with $h^{(\text{gen})}$:

$$x_{t-1} = \sqrt{\alpha_{t-1}}\mathbf{P}_t(\epsilon_t^\theta(x_T^{(1)} | h^{(\text{gen})})) + \mathbf{D}_t(\epsilon_t^\theta(x_T^{(1)})). \tag{2}$$

---

[1]The original DiffStyle work implements a spherical interpolation strategy to produce higher-quality images, but we found this was not necessary for our use case

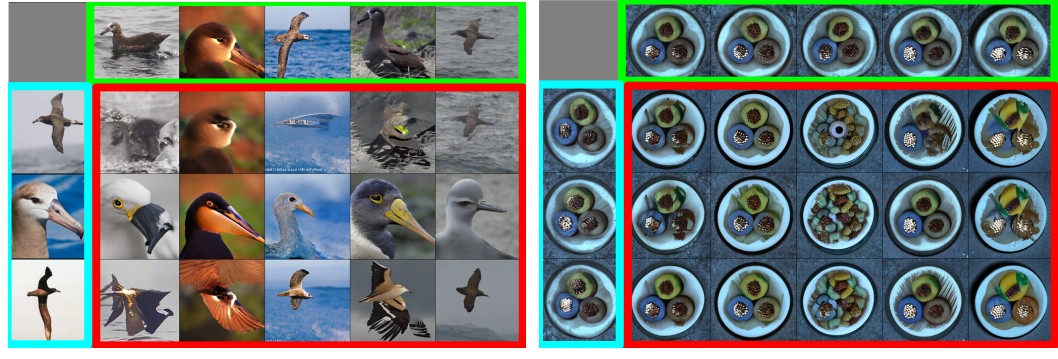

(a) CUB class 1                  (b) MVTec-AD cable

Figure 2: Example pairwise interpolations for CUB class 1 ("Black Footed Albatross") (left) and MVTec-AD product "cable" (right). For each example, the top row of images (in green) are used as source "style" images, and the left column of images (in cyan) are used as source "content" images. The inner grid (in red) shows each pairwise interpolation between the source style and content image, performed with our modified DiffStyle process. All source images are drawn from the distribution of class 1 support images; no validation data or images from other classes are used.

Once the reverse process is completed (i.e., at $t = 0$), the final output $x_0$ is saved as a synthetic anomaly. To generate the full set of synthetic anomalies $\tilde{X}_{out}$, we apply all possibilities of $(I^{(1)}, I^{(2)})$ in the cross product of $X_{style}$ and $X_{content}$. Figure 2 shows examples of anomalies generated with this approach; we find that our generated images have realistic backgrounds and textures, but can contain various semantic corruptions expected of anomalous images. Our synthetic anomaly generation method assumes *no knowledge of the distribution of potential anomalies*: the general-purpose diffusion model is not fine-tuned and only images from the support set $X_{support}$ are used as inputs.

## 4 EMPIRICAL STUDY

To study the efficacy of synthetic anomalies for model selection, we develop an anomaly detection validation benchmark and investigate whether our synthetic validation delivers similar results to ground-truth validation sets. Our evaluation spans various vision domains, including natural and industrial images. We first describe the datasets, anomaly detection tasks, and anomaly generation details in Sec. 4.1. Next, we showcase two use cases of synthetic data in anomaly detection validation—we find that our method selects the true best-performing model in five of six cases (Sec. 4.2) and outperforms all other strategies for CLIP prompt selection (Sec. 4.3); these results are achieved without any access to the real validation data.

### 4.1 EXPERIMENTAL SETUP

We present an experimental setup that can be used as a benchmark to evaluate synthetic validation data. Our benchmark uses a set of anomaly detectors and anomaly detection tasks spanning three vision domains. The tasks vary by difficulty from the easier one-vs-rest to the more difficult one-vs-one anomaly detection setting. The goal of this benchmark is to evaluate how well results on synthetic validation data correspond to results one would obtain with ground-truth validation data; we estimate the absolute detection performance, the relative ranking of anomaly detectors, and the optimal hyper-parameters (such as prompts for CLIP-based anomaly detection).

**Datasets.** Our benchmark spans three frequently-used image datasets: MVTec Anomaly Detection dataset (MVTec-AD) (Bergmann et al., 2019), Caltech-UCSD Birds (CUB) (Wah et al., 2011), and Flowers (Nilsback & Zisserman, 2008); all three datasets were used for the baseline evaluation in (Mirzaei et al., 2023). MVTec-AD contains 15 real industrial product categories; for each category, the training subset contains images of defect-free products, and the testing subset contains labeled images of both good and defective products. CUB and Flowers are multi-class datasets containing 200 bird species and 102 flower species respectively.

**Anomaly detection tasks.** Our benchmark contains 317 anomaly detection tasks: 15 from MVTec-AD, 200 from CUB, and 102 from Flowers, where each class or category is iteratively treated as normal. Besides considering the usual one-vs-rest anomaly detection setup for multi-class datasets CUB and Flowers, we also adopt the one-vs-one setup used in Mirzaei et al. (2023) to simulate more difficult anomaly detection tasks. Specifically, we individually select each class as the inlier class and treat all other images as anomalies—both when considering each out-class individually (one-vs-one) and when considering all out-classes as a single class (one-vs-rest). For MVTec-AD, we predict images of defective products from good products for each product; each product contains multiple defect types, which can be used as different out-classes for one-vs-one measurements or represented as a single class for one-vs-rest. For all tasks, images from the in-class training subset are used as the support set, and images from the relevant in-class and out-class testing subsets are used for constructing the ground-truth validation set. In summary, we report three variants of each task:

- One-Vs-Rest: the AUROC when considering all other classes as a single anomaly class

- One-Vs-One Average: the average AUROC when considering each out-class individually

- One-Vs-One Closest: the worst AUROC when considering each out-class individually

**Generating synthetic anomalies.** For all three datasets and all 317 anomaly detection tasks, we generate synthetic anomalies with training examples from the in-class distribution only. For the CUB and MVTec-AD datasets, we sample 10 images for $X_{\text{style}}$ and 10 images for $X_{\text{content}}$ images from the training set, generating 100 synthetic anomalies. For the Flowers dataset, only 10 images are included in the training set for each class, so we generate 25 synthetic anomalies, using five images for $X_{\text{style}}$ and five images for $X_{\text{content}}$. Figure 2 shows 15 examples of generated synthetic anomalies for a single CUB class (left) and a single MYTec-AD product (right).

Although prior results in Kwon et al. (2023) and Kim et al. (2022) suggest that using a diffusion model trained on the same dataset is required to generate high-quality images, we find that using a single, common diffusion model is sufficient to generate synthetic anomalies. We use the same pre-trained model (the 256x256 diffusion model trained on ImageNet without class conditioning from Dhariwal & Nichol (2021)) for all datasets and anomaly detection tasks.

## 4.2 Model Selection on Synthetic Data

We first demonstrate the effectiveness of our synthetic validation framework for model selection. Given a set of candidate models, we show that one can select the suitable anomaly detection model with the help of our synthetic anomalies generated with DiffStyle.[2]

**Candidate anomaly detection models.** We experiment across five pre-trained ResNet models (ResNet-152, ResNet-101, ResNet-50, ResNet-34, ResNet-18) and five pre-trained Vision Transformers (ViT-H-14, ViT-L-32, ViT-L-16, ViT-B-32, ViT-B-16). We use the ImageNet pre-trained model weights of Dosovitskiy et al. (2021); He et al. (2016). To perform anomaly detection, we follow the methodology of Mirzaei et al. (2023). First, a small set of examples are used to establish a feature bank. Then, to perform detection, each input example is compared to the feature bank: the total Euclidean distance to the 3-nearest neighbors is used as an anomaly score.

**Evaluation setup.** For each task, we vary the number of synthetic anomalies used in the synthetic validation set—from the full set of synthetic anomalies (i.e., $K = 100$ for CUB) to as few as three synthetic anomalies ($K = 3$). When reducing the number of synthetic anomalies, we always use the $K$ anomalies with the lowest anomaly score. We compute the AUROC for each task and candidate model. For each dataset, we average the AUROC across tasks to compute the "synthetic AUROC". We repeat this process with real validation datasets, again averaging over tasks to compute the "real validation AUROC". Lastly, we compare real and synthetic AUROCs to investigate if the rankings of candidate models are similar.

---

[2]We compare with using Tiny-ImageNet as synthetic anomalies for model selection in Appendix A.1. We found that Tiny-ImageNet images do not serve as effective anomalies for model selection.

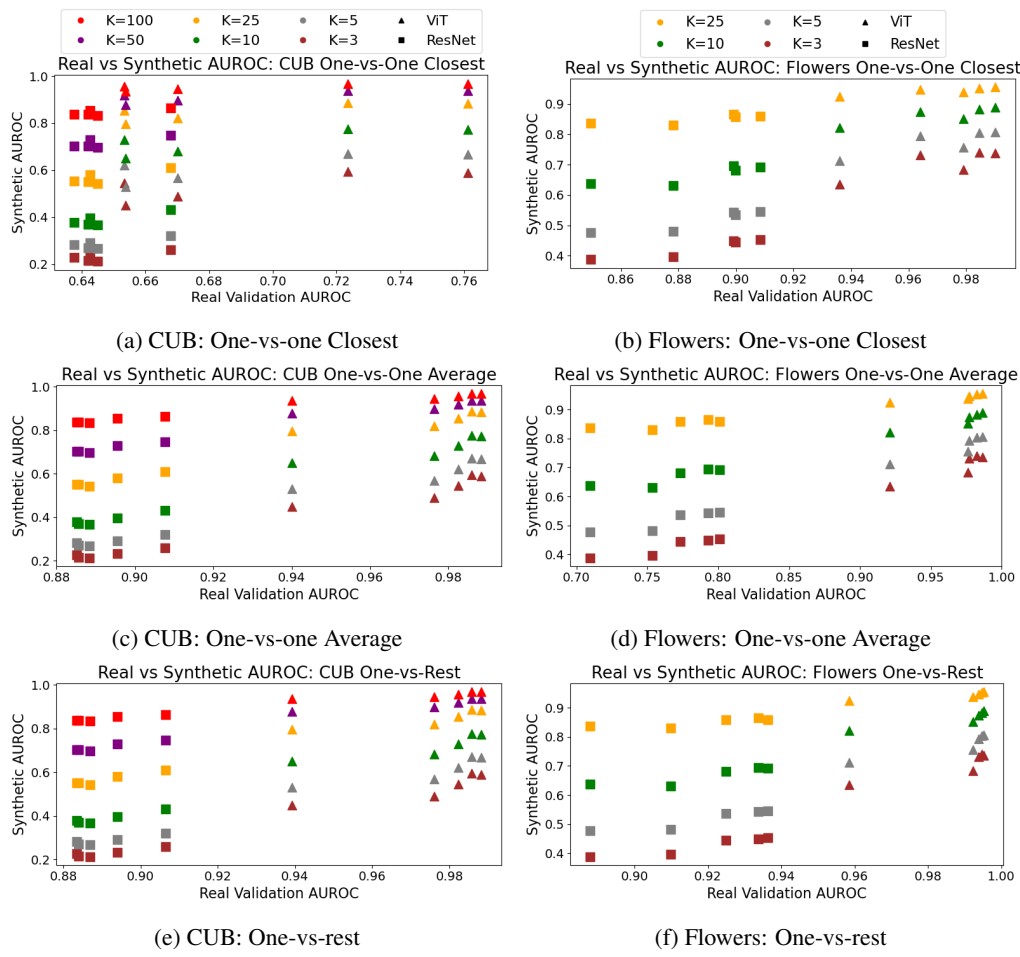

Figure 3: For most anomaly-detection benchmarks, our synthetic validation can be used to select the same best-performing model as with real data. We compare the AUROC with synthetic validation data (y-axis) and real validation data (x-axis) on all three benchmarks: one-vs-one closest (top), one-vs-one average (middle), and one-vs-rest (bottom). Results are shown for the CUB (left) and Flowers (right) datasets. Our synthetic validation is most effective for the one-vs-one average and one-vs-rest benchmarks.

**Evaluation results.** Figure 3 shows the results for the CUB (left) and Flowers (right) datasets. For our synthetic validation to be successful, the model ranking with synthetic data (along the y-axis) should closely match the model ranking with real data (along the x-axis). We find that our synthetic validation best approximate the one-vs-one average and one-vs-rest benchmarks, as they are more robust to variations between classes. Figures 3c–3f show that the rankings of models are consistent: the best-performing model is the same whether real or synthetic validation data is used, and relative rankings of models are similar. We find that the one-vs-one closest benchmark, an approximation of worst-case performance, is more difficult to estimate with synthetic anomalies. Figures 3a and 3b show that, although the real and synthetic AUROCs are relatively aligned for the Flowers dataset, the ranking is less consistent for the one-vs-one benchmark on the CUB dataset.

When performing model selection on the MVTec-AD dataset, we found that our results were ineffective at approximating the real AUROC and selecting the true best models. The results for this experiment are shown in Appendix A.1 and Figure 5. We remark on potential reasons why synthetic validation on the MVTec-AD dataset is more difficult. First, the MVTec-AD dataset presents the fewest tasks (only 15 tasks, compared to 100+ for other datasets), leading to higher variance when averaging. Second, MVTec-AD anomalies are more fine-grained and subtle; Appendix A.2 presents a visualization of the embeddings for all three datasets, and we find that anomalies in the MVTec-

Table 1: Synthetic outliers can effectively select CLIP prompts in the absence of real validation data. We compare five methods for selecting CLIP prompts: (i) our method (described in Sec. 4.3), (ii) a random choice of candidate prompt, (iii) always choosing the default prompt, (iv) our method with Tiny-ImageNet as the synthetic anomalies, and (iv) the single, global best candidate prompt an oracle would provide (grayed out as this cannot be obtained in practice). For each method, we record the tasks where the selected prompt matches the best task-specific prompt, reporting the accuracy and raw count (in parentheses) for the one-vs-one closest and one-vs-one average benchmarks. Across three datasets and two benchmarks, our method outperforms all other strategies, and even outperforms the global best prompt in three out of six cases.

| | | Flowers (102 tasks) | | CUB (200 tasks) | | MVTec-AD (15 tasks) | |
|---|---|---|---|---|---|---|---|
| One-Vs-One Closest | Random Choice | 10.0% | | 10.0% | | 10.0% | |
| | Default Prompt | 0.0% | (1) | 2.5% | (5) | 6.7% | (1) |
| | Tiny-ImageNet Choice | 17.6% | (18) | 17.0% | (34) | 6.7% | (1) |
| | Global Best Prompt (Oracle) | 41.2% | (42) | 18.5% | (37) | 26.7% | (4) |
| | Our Method | **37.2%** | (38) | **23.0%** | (46) | **40.0%** | (6) |
| One-Vs-One Average | Random Choice | 10.0% | | 10.0% | | 10.0% | |
| | Default Prompt | 0.0% | (1) | 0.0% | (0) | 6.7% | (1) |
| | Tiny-ImageNet Choice | 17.6% | (18) | 13.5% | (27) | 6.7% | (1) |
| | Global Best Prompt (Oracle) | 44.1% | (45) | 32.5% | (65) | 40.0% | (6) |
| | Our Method | **40.2%** | (41) | **32.0%** | (64) | **46.7%** | (7) |

Table 2: Using our synthetic outliers to select CLIP prompts always results in the best average AUROC, even in the absence of real validation data. We compare our method for selecting CLIP prompts (described in Sec. 4.3) to the default prompt, the prompt choice made with Tiny-ImageNet, the average of prompts, the hypothetical worst-case (always picking the worst prompt), and the hypothetical best-case (always picking the best prompt). Oracle results are grayed out since they cannot be obtained in practice, but we include them for reference.

| | | Flowers (102 tasks) | CUB (200 tasks) | MVTec-AD (15 tasks) |
|---|---|---|---|---|
| One-Vs-One Closest | Default Prompt | 0.697 | 0.570 | 0.436 |
| | Tiny-ImageNet Choice | 0.718 | 0.582 | 0.438 |
| | Average of Prompts | 0.708 | 0.577 | 0.441 |
| | Worst Prompt (Oracle) | 0.658 | 0.527 | 0.367 |
| | Best Prompt (Oracle) | 0.759 | 0.626 | 0.511 |
| | Our Method | **0.729** | **0.590** | **0.463** |
| One-Vs-One Average | Default Prompt | 0.954 | 0.970 | 0.585 |
| | Tiny-ImageNet Choice | **0.964** | 0.971 | 0.592 |
| | Average of Prompts | 0.957 | 0.971 | 0.582 |
| | Worst Prompt (Oracle) | 0.948 | 0.964 | 0.522 |
| | Best Prompt (Oracle) | 0.968 | 0.976 | 0.646 |
| | Our Method | **0.964** | **0.973** | **0.598** |

AD dataset are the closest to their corresponding benign images. Finally, since our method relies on ImageNet-trained artifacts (ViT, ResNet, CLIP, and our pre-trained diffusion models were all trained with ImageNet) we expect synthetic validation to improve with foundation models that are more representative of the close-up images of industrial products found in the MVTec-AD dataset. However, as shown in the next section, our synthetic validation is still extremely effective for selecting CLIP prompts for all datasets, including MVTec-AD.

## 4.3 CLIP Prompt Selection on Synthetic Data

Zero-shot anomaly detection methods (Jeong et al., 2023b; Liznerski et al., 2022) save the effort on collecting training samples. However, the performance of CLIP-based anomaly detection models depend on the choice of prompts. Prior works evaluate across a variety of candidate CLIP prompts for zero-shot image anomaly detection on real labeled validation data. Validation data might be also absent in the zero-shot setting. Next, we evaluate the efficacy of our generated synthetic anomalies for selecting CLIP prompts on our 317 anomaly detection benchmark tasks.

**Zero-shot anomaly detection with CLIP.** We perform zero-shot image anomaly detection with CLIP (with the ViT-B/16 backbone) as suggested by Liznerski et al. (2022); Jeong et al. (2023b): given an input, we submit two text prompts and predict the class with the higher similarity. We assume that the name of the inlier class is known, and use "`some`" or "`something`" for anomalous classes. For example, for the CUB dataset, if "`red cardinal`" is the name of the inlier class, we compare the CLIP similarities of "`a photo of a red cardinal`" to "`a photo of some bird`". We select amongst a pool of prompt templates from Jeong et al. (2023b) for our anomaly detection tasks. A full list of candidate prompts used for each dataset can be found in Appendix A.5.

**Evaluation setup.** We consider a set of ten candidate prompt templates shown in Appendix A.5.We evaluate each candidate prompt for the one-vs-one closest and one-vs-one average task variants of the 317 anomaly detection tasks and select the prompt with the best AUROC computed on three validation sets: our proposed synthetic validation set, an alternative validation set with anomalies sampled from Tiny-ImageNet, and the ground-truth validation dataset. We measure how often each strategy's selected prompt matches the prompt selected with real validation data. For comparison, we also include two baseline strategies: Random Choice that randomly selects a candidate prompt and Default Prompt that always selects the default prompt template (e.g., "`a photo of a [class name] bird`" vs "`a photo of some bird`"). For reference, as an oracle result, we also report how often a global best prompt matches the task-specific best prompts.

**Evaluation results.** For all three datasets, we report the prompt selection accuracy in Table 1 and averaged AUROCs for different prompt-selection strategies in Table 2. In Table 1, compared to all baselines, our method performs best for all tasks, outperforming all other strategies by 6–40%. This suggests that our generated synthetic anomalies are helpful in selecting the best prompts for tuning CLIP-based anomaly detectors on various tasks. Furthermore, our method outperforms the single-best-prompt strategy on in three out of six settings (two task variants in three datasets shown in Table 1). In these cases, the prompt selection accuracy difference ranges from -4% to +16%. We emphasize that selecting the single best-performing prompt relies on access to the real validation data, whereas our method only uses synthetic anomalies generated from in-class training sources.

As shown in Table 2, by selecting better prompts with our generated synthetic anomalies, we improve the zero-shot anomaly detection results in the challenging one-vs-one closest setting over the popular default choice of prompt, i.e., "`a photo of a [class name]`" by 3.4% on Flowers, 2% on CUB, and 2.7% on MVTec-AD. Our proposed prompt selection also provides a consistent performance improvement over the averaged results of various prompts and the results of prompts selected with Tiny-ImageNet, showcasing the general effectiveness of our method.

## 5 Conclusion

In this work, we investigate an approach for anomaly detection model selection and validation *without any validation data*. To substitute for a validation dataset, we propose that a general-purpose diffusion model can be used generate synthetic outliers, using only a small support set of in-class examples as input. Unlike prior methods for synthetic anomaly generation, we do not rely on any model training, fine-tuning, or any domain-specific architectures or techniques. Our empirical study shows that synthetic validation datasets constructed with this approach can be used effectively in two ways: for selecting amongst a set of candidate anomaly detection models, and for selecting hyper-parameters for zero-shot anomaly detection models, specifically CLIP prompt templates.

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

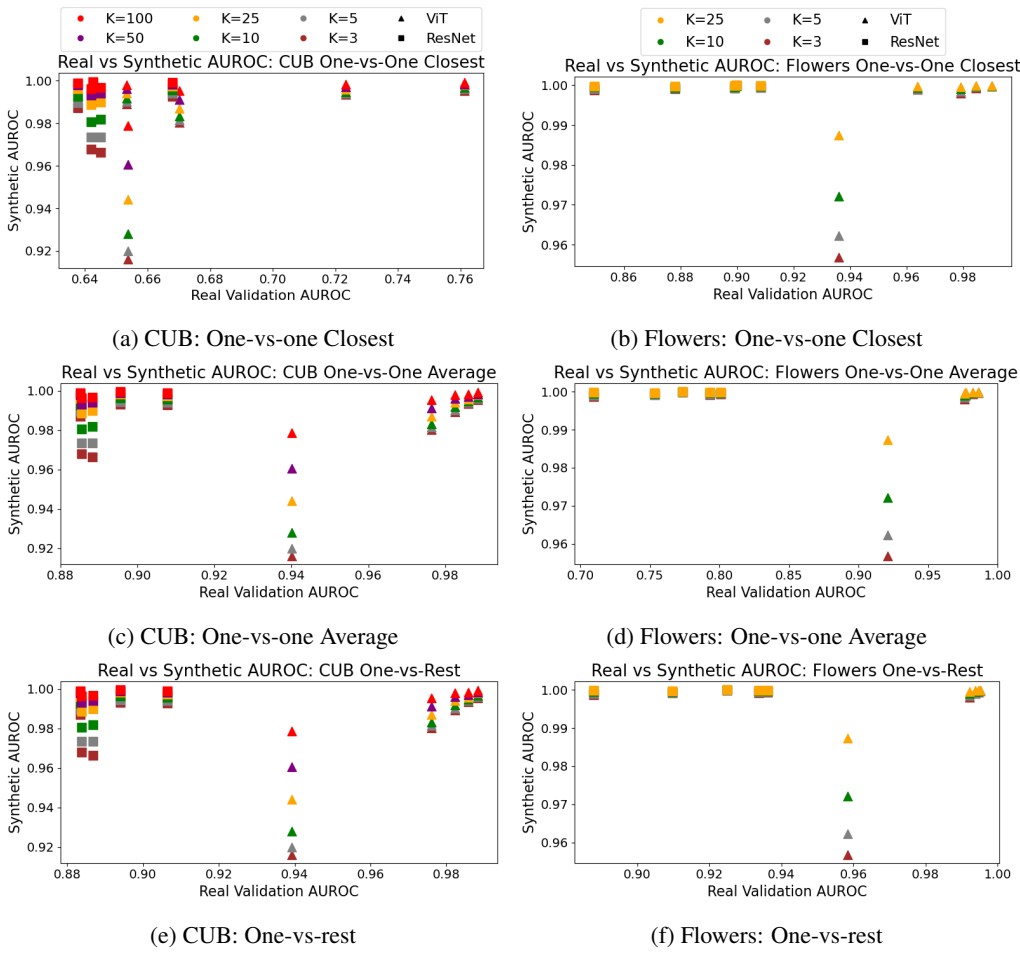

(a) CUB: One-vs-one Closest        (b) Flowers: One-vs-one Closest

(c) CUB: One-vs-one Average        (d) Flowers: One-vs-one Average

(e) CUB: One-vs-rest        (f) Flowers: One-vs-rest

Figure 4: We compare the real validation AUROC against the AUROC when Tiny-Imagenet is used as synthetic anomalies, for the CUB (left) and Flowers (right) datasets and for all three benchmarks. We find that the performance is poor: model selection cannot be reliably performed when Tiny-ImageNet examples as used as synthetic anomalies.

# A  APPENDIX

## A.1  ADDITIONAL RESULTS

**Model selection with Tiny-ImageNet.**  Our initial experiments investigated if Tiny-ImageNet could be used effectively as synthetic anomalies when constructing the synthetic validation set. For these experiments, the same support set for each anomaly detection task is the same as the support for the corresponding experiment with our generated anomalies. When sampling anomalies from Tiny-Imagenet, we sample uniformly at random to generate a dataset $\tilde{X}_{\text{out}}$ of the same size: 100 images for tasks with the CUB and MVTec-AD datasets, and 25 images for tasks with the Flowers dataset. Ultimately, we found that using Tiny-ImageNet examples were not effective for our chosen tasks; in addition to the results for CLIP prompt selection in Table 1, the results for model selection are shown in Figure 4.

**Additional MVTec-AD results.**  Figure 5 shows the results of the model selection experiment with the MVTec-AD dataset on the one-vs-one average and the one-vs-rest benchmarks (which were shown to be easier benchmarks to estimate in Figure 3). Unlike the CUB and Flowers datasets, in which synthetic anomalies could successfully approximate real validation performance, our synthetic anomalies are less effective for MVTec-AD.

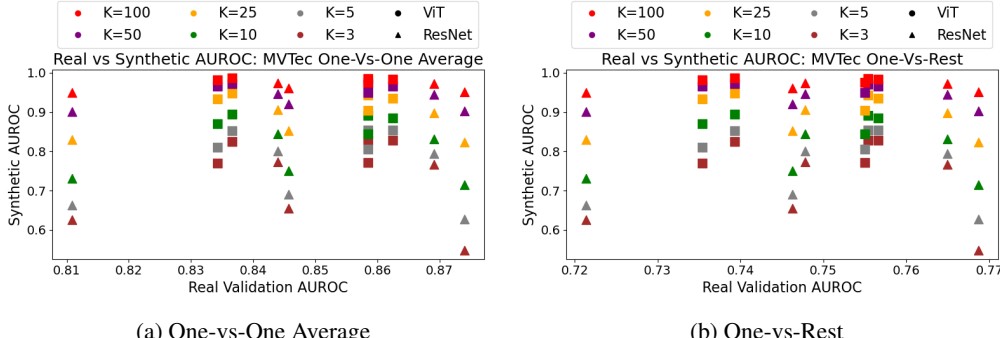

(a) One-vs-One Average            (b) One-vs-Rest

Figure 5: Comparing the real validation AUROC against the synthetic validation AUROC for the MVTec-AD dataset. For all the one-vs-one average and the one-vs-rest benchmark, using our synthetic validation dataset is ineffective for selecting the true, best anomaly detection models.

We also comment on the performance of our method when selecting CLIP prompts for MVTec-AD. We noticed a disparity in performance between objects in MVTec-AD (e.g., capsules, cables, or screws) and textures in MVTec-AD (e.g., carpets, wood, or tiles). Our method is unable to select the best CLIP prompt in any of the six textures for MVTec-AD, instead only performing well on the nine MVTec-AD objects. We therefore identify yet another challenge when using our approach for MVTec-AD— DiffStyle relies on assumptions of "style" and "content" in their source images, and these elements are not present in MVTec-AD textures like carpets or tiles.

## A.2 Visualizing Differences between Anomaly Detection Tasks

In this section, we visualize the differences in anomaly detection tasks between the Flowers, CUB, and MVTec-AD datasets. In doing so, we provide insight into situations when our method may or may not perform well. Figure 6 shows the t-SNE visualizations for four tasks classes from each dataset, using the embedding space of the ViT-B-16 vision transformer. For the Flowers and CUB datasets, we select the "abnormal" class as the one that produces the lowest AUROC in a one-vs-one setting (i.e., one-vs-one closest[3]). We visualize the (i) normal class (in orange), (ii) abnormal class (in blue), and (iii) synthetic validation (in black) for each task.

For the Flowers dataset (top row of Figure 6), real abnormal data and synthetic abnormal data are approximately similar in distribution, but are easy to distinguish from real normal data. For the CUB dataset (middle row of Figure 6), real abnormal data and synthetic abnormal data are both approximately equal in distance from the real normal class; classes in the CUB dataset are more difficult to distinguish between each other, and our synthetic validation dataset is approximately as difficult to distinguish. Finally, for the MVTec-AD dataset (bottom row of Figure 6), for which benchmarks were the hardest to estimate with synthetic validation data, the real normal data and real abnormal data are much closer in distance. Our synthetic validation data is unable to approximate the areas that lie between real and abnormal data in the MVTec-AD dataset.

## A.3 Generating Synthetic Outliers with Text-guidance

A directional CLIP loss is defined using CLIP's image encoder $E_I$, CLIP's text encoder $E_T$, and source-target image and text pairs $(x_{source}, x_{target})$ and $(y_{source}, y_{target})$ respectively, StyleGAN-NADA enables text-guided generation of images:

$$\Delta T = E_T(y_{target}) - E_T(y_{source})$$
$$\Delta I = E_I(x_{target}) - E_I(x_{source})$$
$$L_{dir} = 1 - \frac{\Delta I \cdot \Delta T}{||\Delta I||||\Delta T||}$$

---

[3]We do not visualize cases where the lowest AUROC was 1, indicating perfect detection across all classes.

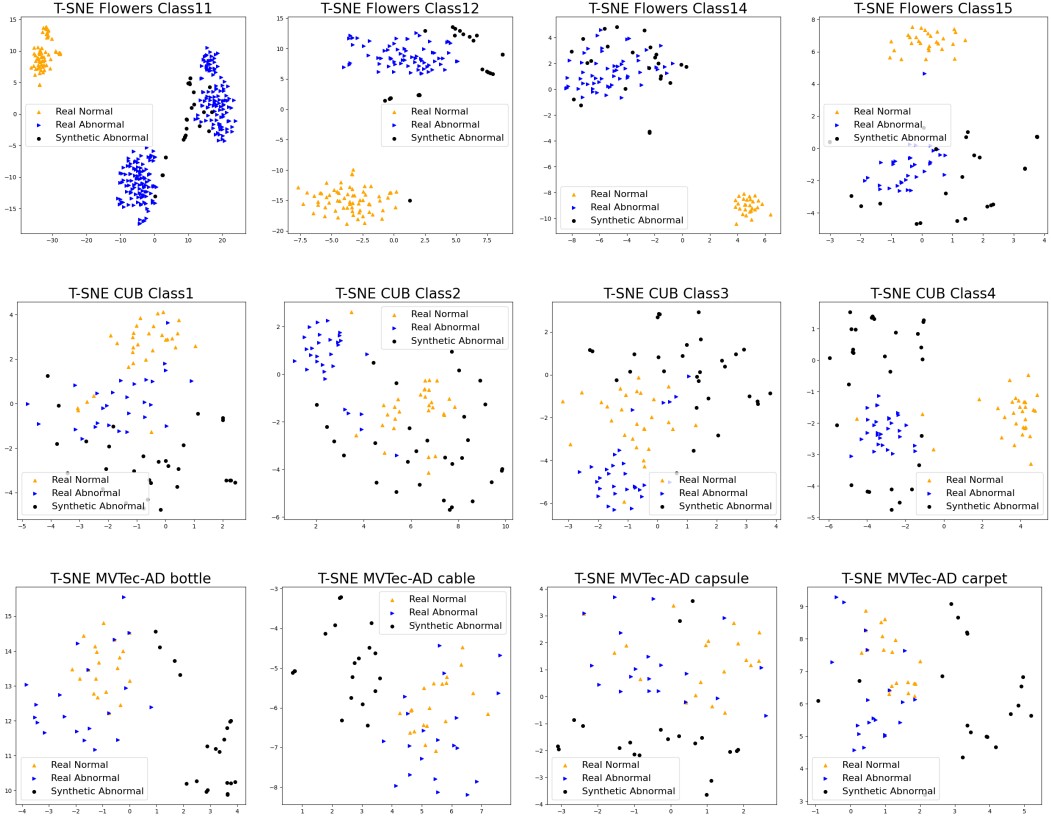

Figure 6: For each anomaly detection task, we present the t-SNE visualizations of the embedding space of the ViT-B-16 vision transformer. We visualize four tasks from the Flowers dataset (top row), CUB dataset (middle row) and the MVTec-AD dataset (bottom row), comparing the real normal data, real abnormal data, and synthetic abnormal data for each task. We find that our synthetic abnormal data is able to approximate the distances and distributions observed in real abnormal data for the Flowers and CUB datasets. However, for the MVTec-AD dataset, the differences between real normal data and real abnormal data are much smaller, and our synthetic abnormal data is unable to approximate it.

We use the Asyrp process (Kwon et al., 2023) for text-guided anomaly generation, but modify Asyrp in two ways: (i) using non-domain-specific text-guidance and (ii) defining the edit-strength of each anomaly. The original Asyrp process is evaluated on well-defined domains, and assumes that the source and target text are known (e.g, modifying "face" to "smiling face"). We instead propose a methox that does not assume a specific domain and does not require domain-specific texts as input.

First, we find that using a source text is unneeded, and a meaningful direction for $\Delta T$ can be extracted by using the image encoder $E_I$ and the source image $x_{source}$. Second, we find that target texts can be replaced with auxiliary, out-of-domain texts, which we call $y_{aux}$. We make these two changes to redefine our directional loss $L'_{dir}$.

$$\Delta T' = E_T(y_{aux}) - E_I(x_{source})$$
$$\Delta I = E_I(x_{target}) - E_I(x_{source})$$
$$L'_{dir} = 1 - \frac{\Delta I \cdot \Delta T'}{||\Delta I||||\Delta T'||}$$

We follow the training procedure for h-space extractor defined in Kwon et al. (2023) to extract $\Delta h$, the direction in h-space that creates the desired change. We then apply Asyrp during the reverse diffusion process, but use the linear property of h-space and define an edit strength $\alpha$, which indicates

how strongly we modify the image. Referring to the formulation defined in Sec. 3.2, we modify the reverse DDIM process on the latent vector $x_T^{(1)}$ by adding our h-space term:

$$x_{t-1} = \sqrt{\alpha_{t-1}}\mathbf{P}_t(\epsilon_t^\theta(x_T^{(1)}|h^{(1)} + \alpha\Delta h)) + \mathbf{D}_t(\epsilon_t^\theta(x_T^{(1)})). \tag{3}$$

## A.4 USING SYNTHETIC OUTLIERS FOR OUTLIER EXPOSURE

In addition to using synthetic outliers for validation, synthetic outliers can also be used for improving the performance of anomaly detection models through outlier exposure. Our methodology closely follows that of Fort et al. (2021); Mirzaei et al. (2023): we use a pre-trained vision transformer model, fine-tune the vision transformer on a surrogate classification task, and use distances in the trained embedding space as an anomaly detection.

Mirzaei et al. (2023) use a surrogate classification task for fine-tuning—a binary classification layer is added to the vision transformer, and the model is trained on benign in-class examples and synthetic outlier examples. In addition to the surrogate classification task, we also propose a regression-based task. We generate a variety of synthetic outliers with text-guidance, using varying edit strength $\alpha$. When fine-tuning anomaly-detection models, the surrogate task is a regression that predicts $\alpha$.

After fine-tuning, we remove the prediction head of the vision transformer. The support set is then converted into the transformer's embedding space (i.e., the last layer before the prediction layer) and used as a feature bank for anomaly detection. At test time, the total Euclidean distance to the closest three examples in the feature bank is used as the anomaly score.

## A.5 CLIP PROMPT TEMPLATES

For our experiments in Sec. 4.3, we evaluated across set of ten candidate prompt templates. Our evaluated prompts are general-purpose, and only the term "bird" or "flower" is added to the template for the CUB and Flowers dataset respectively. For each dataset, the candidate prompt templates are provided below:

```
% CLIP Templates for Flowers
['a photo of a {} flower', 'a photo of some flower'],
['a cropped photo of a {} flower', 'a cropped photo of some flower'],
['a dark photo of a {} flower', 'a dark photo of some flower'],
['a photo of a {} flower for inspection', 'a photo of some flower for inspection'],
['a photo of a {} flower for viewing', 'a photo of some flower for viewing'],
['a bright photo of a {} flower', 'a bright photo of some flower'],
['a close-up photo of a {} flower', 'a close-up photo of some flower'],
['a blurry photo of a {} flower', 'a blurry photo of some flower'],
['a photo of a small {} flower', 'a photo of a small some flower'],
['a photo of a large {} flower', 'a photo of a large some flower'],
```

```
% CLIP Templates for CUB
['a photo of a {} bird', 'a photo of some bird'],
['a cropped photo of a {} bird', 'a cropped photo of some bird'],
['a dark photo of a {} bird', 'a dark photo of some bird'],
['a photo of a {} bird for inspection', 'a photo of some bird for inspection'],
['a photo of a {} bird for viewing', 'a photo of some bird for viewing'],
['a bright photo of a {} bird', 'a bright photo of some bird'],
['a close-up photo of a {} bird', 'a close-up photo of some bird'],
['a blurry photo of a {} bird', 'a blurry photo of some bird'],
['a photo of a small {} bird', 'a photo of a small some bird'],
['a photo of a large {} bird', 'a photo of a large some bird'],
```

```
% CLIP Templates for MVTec
['a photo of a {}', 'a photo of something'],
['a cropped photo of a {}', 'a cropped photo of something'],
['a dark photo of a {}', 'a dark photo of something'],
['a photo of a {} for inspection', 'a photo of something for inspection'],
['a photo of a {} for viewing', 'a photo of something for viewing'],
['a bright photo of a {}', 'a bright photo of something'],
['a close-up photo of a {}', 'a close-up photo of something'],
['a blurry photo of a {}', 'a blurry photo of something'],
['a photo of a small {}', 'a photo of a small something'],
['a photo of a large {}', 'a photo of a large something'],
```

