# OpenReview forum: "Model Selection of Anomaly Detectors in the Absence of Labeled Validation Data"
_ICLR.cc/2024/Conference — Submitted to ICLR 2024_

### Official Review · Reviewer_Wc92 · 2023-10-29

**Soundness:** 3 good
**Presentation:** 3 good
**Contribution:** 3 good
**Rating:** 5
**Confidence:** 5

**Summary:**

To address the problem of sparse annotated data in existing detection tasks, the paper designed a framework that uses Diffusion Model interpolation to generate abnormal data, and then uses the synthesized data to perform model selection for anomaly detection. The paper conducted extensive experiments on both natural image data and industrial image data, demonstrating the effectiveness of this framework.

**Strengths:**

+ To address the problem of sparse annotated samples, the paper uses interpolation with Diffusion Model to transform normal images into abnormal images with certain semantic information, thus simulating some abnormal data well.
+ The paper validated the effectiveness of synthesized data through extensive experiments on different datasets and models.
+ The paper validated the effectiveness of synthesized data in selecting prompts for zero-shot detection using CLIP.

**Weaknesses:**

+ Although the paper's method has significant effects on the Flowers and CUB datasets, it does not perform well on the MVTec AD dataset. From Tables 1 and 2, it can be seen that the synthesized data is not helpful for the MVTec dataset. From Figure 2, it can also be seen that the interpolation synthesis has poor performance. For the CUB dataset, the anomalies are more significant, so the synthesized data is effective, but for the MVTec dataset, the anomalies are more subtle, so the synthesized data is not effective.
+ The paper's abnormal synthesis function is entirely based on DiffStyle, and it remains to be verified whether the method of interpolating abnormal images from different normal images is reasonable. Perhaps perturbing features in different dimensions in the latent space may have better results. The paper should consider different designs for high-level semantic anomalies and low-level semantic anomalies in this regard.

**Questions:**

+ It is unclear what considerations the authors had in comparing the Flowers, CUB, and MVTec datasets. There are significant differences between these datasets.
+ The paper's workload is significant, but the experimental character is too strong, and it is unclear whether the authors have any ideas for redesigning the Diffusion generation process based on the experimental results.

---

> ### Author Response · Authors · 2023-11-22
> **Response to reviewer Wc92**
>
> Thank you for your comments.
>
> We selected the CUB, Flowers, and MVTec-AD datasets as they were the baseline datasets used in Mirzaei et al (2023), see Table 1 in their work. Mirzaei et al. also use CIFAR-10 and CIFAR-100, but we found that the images in these datasets were of too low resolution for the ImageNet-pretrained diffusion model, and upsampled versions of these images did not produce good results with DiffStyle.
>
> There are many options for modifying how the diffusion process works. The first, and simplest ideas would be to change (i) what images are used as “style” and “content” for DiffStyle and (ii) to modify the value of gamma in the DiffStyle generation process (which modifies the style-content proportions). The original DiffStyle paper (Jeong et al. 2023a) evaluates both of these parameters. Rather than search over the parameter space for optimal results, our paper shows that, even with a simple combination of images and default DiffStyle values, our method is effective on some (i.e., CUB and Flowers) datasets for model selection! This is a valuable result on its own and follow-up experiments would be interesting, but are beyond the scope of the current paper.

---

> > ### Comment · Reviewer_Wc92 · 2023-11-23
> >
> > After reading the paper *Fake It Until You Make It: Towards Accurate Near-Distribution Novelty Detection*, I remain skeptical about the choice of dataset. If meaningful detection cannot be performed on complex datasets, then the results should not be discussed as valid contributions. The paper's interpolation generation method does have some impact on certain natural image datasets, but it does not significantly differ from previous works in this regard.

---

### Official Review · Reviewer_tN6d · 2023-10-31

**Soundness:** 4 excellent
**Presentation:** 4 excellent
**Contribution:** 3 good
**Rating:** 6
**Confidence:** 4

**Summary:**

In the paper "Model Selection of Anomaly Detectors in the Absence of Labeled Validation Data" the authors consider the task of anomaly detection in a semi-supervised setting where only normal data is given for training. For selecting a suitable anomaly detector, the authors propose to augment the validation data by anomalous data points that are created with the help of diffusion models. In their empirical study, the authors find the synthetically created anomalies to give rise to a good choice of anomaly detectors.

**Strengths:**

- Novel method and intriguing idea to create synthetic anomalies for images as an input.
- Strong empirical performance of the proposed method
- Thorough evaluation of the method and good set of baselines.
- The related literature is nicely reviewed and the overall presentation is excellent.

**Weaknesses:**

- The authors could elaborate more on the limitations of the approach. For instance, one problem I see is that the synthetically generated anomalies do not necessarily resemble the ground truth distribution of anomalies. In particular, the question for the image classification datasets is indeed what would an actual anomaly look like? In particular, to me, it is questionable whether the generated images make sense at all as the observations would probably never be made in the real world.
- Figure 3 needs more explanation what exactly is plotted. The legend is of the figure is also off, e.g., in Figure 3(a) there are squares which are not contained in the legend.

**Questions:**

- How are anomalies even defined in the image domain? Is it just out of distribution data? Unrealistic images? If the latter, why would one even expect to observe such images?

---

> ### Author Response · Authors · 2023-11-22
> **Response to reviewer tN6d**
>
> Thank you for your comments. In our updated PDF, we have corrected the legend, caption, and description around Figure 3.
>
> Regarding the nature of anomalies, we do not claim that our synthetic anomalies are indicative of real anomalies; we only claim that they can be used to estimate the performance one would get with real validation data. If our method were able to produce realistic anomalies, then these examples could be added to real validation datasets.
>
> This highlights the importance of our technique; since we assume a setting where real anomalies are difficult to generate and cannot be reproduced, there is a strong need for synthetic data that can be used to estimate performance, even if they do not visually represent the distribution of anomalies.

---

> > ### Comment · Reviewer_tN6d · 2023-11-22
> > **Re: Response to reviewer tN6d**
> >
> > For me the two claims of obtaining a truthful performance estimate via generation of synthetic anomalies and generating synthetic anomalies thar are indicative of real anomalies. How could the performance estimate be truthful if not computed for the actual distribution of anomalies?

---

> ### Author Response · Authors · 2023-11-22
>
> Thank you for your follow up question!
>
> It is possible to estimate the performance on actual anomalies using the performance on synthetic data, provided that the distributions are sufficiently close. We show this result empirically, but for a theoretical result, see Section 3 of this recent paper at ICML 2023  (https://proceedings.mlr.press/v202/shoshan23a/shoshan23a.pdf), which also considers synthetic data for model selection. From Corollary 3.3: "if the total variation between the real and synthetic distributions is not larger than the synthetic risk difference between a pair of hypotheses, then their error ranking is preserved across domains".
>
> Note that our work makes far fewer assumptions than that of Shoshan et al. (2023). Their work trains a GAN to generate synthetic data for model selection in each domain, which is a very expensive process. In contrast, our work performs model selection without any training or fine-tuning of a generative model.

---

### Official Review · Reviewer_v3es · 2023-11-04

**Soundness:** 2 fair
**Presentation:** 2 fair
**Contribution:** 2 fair
**Rating:** 6
**Confidence:** 3

**Summary:**

The authors propose to generate synthetic outlier data for outlier detection tasks. The method is based from mapping images into latents, then taking a mixture in latent space and mapping it back using a diffusion model.
The evaluate the performance of one outlier detection method in multiple setups and 3 datasets. Furthermore they evaluate the suitability of the generated outliers for selecting prompts for the usage of CLIP as foundational model for zero-shot outlier detection.

**Strengths:**

They ask an important question, namely how reliable is synthetic outlier data for the evaluation of outlier detection setups.
The idea is clear and simple.

**Weaknesses:**

I think the question would need to be evaluated for more outlier detection methods, not just one distance based one.

Also the method to generate outliers is very simple.
It is a simplified version of mixup in latent space. It offers no control over what kind of outliers are created.
Calling it a style and content mixture is dubious, because the method seemingly has no attempted separation into content and style. it seemingly has only 1 latent space.
This would also benefit from a more thorough evaluation of creating outliers. e.g. stochastic mixup in latent space, or inpainting, and so on.

Also it is not clear to what extent the method learns to discriminate real image properties from synthetic ones - this is because the real images are never run through the encoding-decoding step.

This conclusion is not true:
In an extensive empirical study, ranging from natural images to industrial applications, we find that our synthetic validation framework selects the same models and hyper-parameters as selection with a ground-truth validation set.

yes in the simple class vs other classes on flowers and birds it holds, on the more realistic MVTec it does not hold, see their appendix.

It is not bad per se, if the proposed method does not work, but putting a questionable conclusion in the abstract is misleading.

The zero-shot task result is interesting scientifically. Practically it is unlikely that one would use that for serious outlier detection tasks.

**Questions:**

What are the five repetitions in Figure 3? Can they be compared against each other ?

How would the outlier detection perform for reconstruction based AD methods ? or maybe another class ?

Can experiments be run to ascertain the usefulness for smaller defects beyond MVTec ? What if the outliers are not on semantic level but more on imaging setting differences ?

Can an experiment be performed to understand to what extent the method classifies real vs diffusion generated images ?

---

> ### Author Response · Authors · 2023-11-22
> **Response to reviewer v3es**
>
> We have followed your suggestion and have modified our abstract and introduction to better reflect the conclusions of our empirical study. “​​Our synthetic validation framework selects the same models and hyper-parameters as selection with a ground-truth validation set when evaluated on natural images”.
>
> We would like to clarify any potential misunderstandings about our work.
> - “Style” and “content” is dubious: We borrow the terms style and content from Jeong et al. 2023a. Note that in DiffStyle, the original image used in the forward pass (the style image) maintains its skip connection values during the backward pass (when the content is injected), so DiffStyle is not symmetric: DiffStyle(x,y) != DiffStyle(y,x)
> - What Figure 3 is showing: We evaluated over five pretrained ResNet models and five pretrained vision transformers (for a total of 10 datapoints per setting). While these settings can be compared to each other (to determine which model is best), what is more important is the relative rankings of these models. Figure three shows that, amongst the ten candidate models, the model with the best performance on our synthetic validation set matches the best performing model on the real validation set.
> - Classifying real vs diffusion-generated images: Figure 3 shows these results. Values on the x-axis are the real-vs-real AUROC values, and values on the y-axis are the real-vs-diffusion-generated AUROC values. We find that the real-vs-diffusion-generated AUROC can be used to select the same models as the real AUROC.

---

> > ### Comment · Reviewer_v3es · 2023-11-23
> >
> > Can an experiment be performed to understand to what extent the method classifies real vs diffusion generated images ?
> >
> > Your reply to this might be based on a mutual misunderstanding. My intent was to ask: if one trains on synthetic images as anomalies mixed with normal unmodified images, it may learn to discern synthetic images vs real images.  The question was, to what extent does this happen ?

---

> > > ### Comment · Reviewer_v3es · 2023-11-23
> > >
> > > After rebuttal, the reviewers is raising the score to marginally above. What is missing for a higher score are more experiments about how to create outliers and a more broad evaluation in this regard.

---

### Author Response · Authors · 2023-11-22
**Response to all authors**

We thank the reviewers for their insightful comments and suggestions! Based on responses commonly shared between reviewers, we have updated our submission PDF to include the following:
- We have updated the performance claims in the abstract: “​​our synthetic validation framework selects the same models and hyper-parameters as selection with a ground-truth validation set when evaluated on natural images”, as opposed to all datasets.
- We have corrected the legend in Figure 3, and improved the writing in the caption and main text body to make it clearer that it visualizes the comparison between real AUROC (on real validation data) and synthetic AUROC (on our generated anomalies). We have also added the description of K, which is the number of synthetic examples used for validation.
- We have included a reference to Mirzeai et al. (2023), explaining that we chose our three datasets based on their baseline evaluation.
- In our updated PDF, we include Appendix A.2, which contains visualizations of the embedding space for the CUB, Flowers, and MVTec-AD datasets. These visualizations provide some insight into why our method works best on CUB/Flowers, but has more difficulty on MVTec-AD; the differences between normal images and anomalies in MVTec-AD are the smallest and most difficult to approximate with synthetic anomalies. We conclude that our approach is likely to be more effective on natural variations (like in CUB/Flowers) as opposed to fine-grained anomalies (like in MVTec).

Multiple reviewers have suggested that we evaluate with other anomaly detection methods and other anomaly generation methods. We agree that different anomaly types might require different synthesis styles. The main goal of our paper is to demonstrate that synthetic anomalies can be used for AD validation and model selection. We demonstrate one way of generating such synthetic anomalies, and study it on the same datasets as Mizreai et al ‘23. While we agree with the reviewers that trying more generation techniques will be an interesting research endeavor, we defer it to future work.. The main focus of the work presented in this paper, is to propose a new framework for synthetic validation, with a new way of thinking about AD model selection and a new benchmark setting for evaluating different generation strategies.

Finally, we would like to re-emphasize the limitations of our setting, and why model selection without any validation data is so difficult: we assume a small support set of only benign images, no validation data, no model finetuning, and no domain-specific techniques. These assumptions are limiting, yet many practical settings match them: it is often too expensive for common practitioners to train new diffusion models, finetune existing anomaly detectors, or design specific architectures for their tasks, but they often have some benign examples and want to select amongst a set of public pre-trained architectures.

Our method is exactly compatible with these assumptions. Despite several limitations, our method is able to effectively perform model selection on natural images and CLIP prompt select on all evaluated datasets, using only public, pre-trained artifacts! We believe that our work will trigger interest for the community to also explore other strategies for anomaly generation when used for model selection, and that our initial results in the face of several practical yet difficult assumptions encourages more work that tackles this challenging problem!

Thank you again for your constructive feedback.

---

### Meta-Review · Area_Chair_yfJt · 2023-12-09

**Metareview:**

The paper presents an innovative approach to anomaly detection, focusing on the use of synthetic anomalies for model selection in the absence of labeled validation data. The authors propose a novel framework that circumvents the need for training, fine-tuning, or domain-specific architectures, demonstrating its effectiveness particularly on natural images using pre-trained artifacts.

The methodology has been acknowledged for its clear concept and practical assumptions, meeting a critical need in the field. However, the paper faces criticism for its limited evaluation scope, primarily focusing on a single outlier detection method. Additionally, concerns have been raised about the realism of the generated synthetic anomalies and their applicability to complex datasets like MVTec-AD, which could limit the generalizability of the findings.

**Justification For Why Not Higher Score:**

The negative feedback is primarily on the paper's limited evaluation range and the potential lack of realism in its synthetic anomalies. The method, while innovative, is tested mainly on one type of outlier detection approach, limiting the understanding of its broader applicability.

Additionally, the effectiveness of the synthetic anomalies in accurately representing real-world scenarios remains a concern after the discussion phase (particularly highlighted in their performance on more complex datasets like MVTec-AD). These limitations suggest the need for a more thorough exploration and validation of the methodology across varied contexts.

**Justification For Why Not Lower Score:**

The paper's score should not be lower because it addresses a crucial and challenging aspect of anomaly detection with a novel and practical approach. The paper successfully introduces a new perspective; the method’s compatibility with practical constraints and its promising results on certain datasets underscore its potential and relevance.

---

### Decision · Program_Chairs · 2024-01-16

Reject